# Oligomeric scaffolding for curvature generation by ER tubule-forming proteins

Yun Xiang [1,2], Rui Lyu[1,2] & Junjie Hu [1,2] ✉

The reticulons and receptor expression-enhancing proteins (REEPs) in the endoplasmic reticulum (ER) are necessary and sufficient for generating ER tubules. However, the mechanism of curvature generation remains elusive. Here, we systematically analyze components of the REEP family based on AI-predicted structures. In yeast REEP Yop1p, TM1/2 and TM3/4 form hairpins and TM2-4 exist as a bundle. Site-directed cross-linking reveals that TM2 and TM4 individually mediate homotypic dimerization, allowing further assembly into a curved shape. Truncated Yop1p lacking TM1 (equivalent to REEP1) retains the curvature-generating capability, undermining the role of the intrinsic wedge. Unexpectedly, both REEP1 and REEP5 fail to replace Yop1p in the maintenance of ER morphology, mostly due to a subtle difference in oligomerization tendency, which involves not only the TM domains, but also the TM-connecting cytosolic loop and previously neglected C-terminal helix. Several hereditary spastic paraplegia-causing mutations in REEP1 appear at the oligomeric interfaces identified here, suggesting compromised self-association of REEP as a pathogenic mechanism. These results indicate that membrane curvature stabilization by integral membrane proteins is dominantly achieved by curved, oligomeric scaffolding.

The endoplasmic reticulum (ER) is composed of interconnected sheets and tubules[1,2]. ER tubules are cylindrical structures with a relatively constant diameter of ~30 nm in yeast and ~50 nm in mammalian cells[3,4]. Compared to cisternal sheets, tubules have characteristic membrane bending in their cross-section, which is similar to the edge of sheets. These curved membranes are stabilized by the same class of integral membrane proteins, known as the reticulons (RTNs) and receptor expression-enhancing proteins (REEPs)[5]. RTNs and REEPs are enriched in the ER tubular network but rarely appear in sheets and the nuclear envelope[5]. Therefore, they are commonly used as a reliable marker of tubular ER. Deletion of Rtn1p and Yop1p, the yeast REEP, in yeast cells causes loss of tubules and aberrant ER morphology[5]. Overexpression of RTN or REEP in mammalian cells generates abundant tubule bundles[6]. Purified Yop1p or Rtn1p, when reconstituted into proteoliposomes, forms membrane tubules in vitro[7]. Collectively, RTNs and REEPs are essential for generating ER tubules. As expected, these proteins also localize to the edge of ER sheets and stabilize the membrane curvature[8].

ER tubule-forming proteins are evolutionarily conserved[9]. Yeast has two RTNs (Rtn1p and Rtn2p) and one REEP (Yop1p, most similar to REEP5)[5]. Apicomplexa parasite *Plasmodium berghei* possesses one RTN homolog (*Pb*RTN1), one REEP1 (*Pb*YOP1L), and one REEP5 (*Pb*YOP1)[10,11]. Mammals contain four RTNs and six REEPs[9]. Among different REEPs, REEP1-4 and REEP5/6 form two subgroups[12]. The common feature of all of these proteins is the reticulon homology domain (RHD), which consists of two consecutive transmembrane (TM) hairpins (Fig. 1a). Each pair has 30–35 residues, which are too long to pass the membrane only once but too short to pass it twice[5]. The RHD has subsequently been found in several other ER-resident proteins, including the Pex30/MCTP2 family, ARL6IP1, EI24, and Atg40/FAM134B, some of which also contribute to ER tubule formation[13–17].

Mutations in RTN2 and REEP1 are linked to hereditary spastic paraplegia (HSP)[18,19], which is characterized by progressive spasticity

[1]National Laboratory of Biomacromolecules, CAS Center for Excellence in Biomacromolecules, Institute of Biophysics, Chinese Academy of Sciences, Beijing 100101, China. [2]University of Chinese Academy of Sciences, Beijing 100101, China. ✉e-mail: huj@ibp.ac.cn

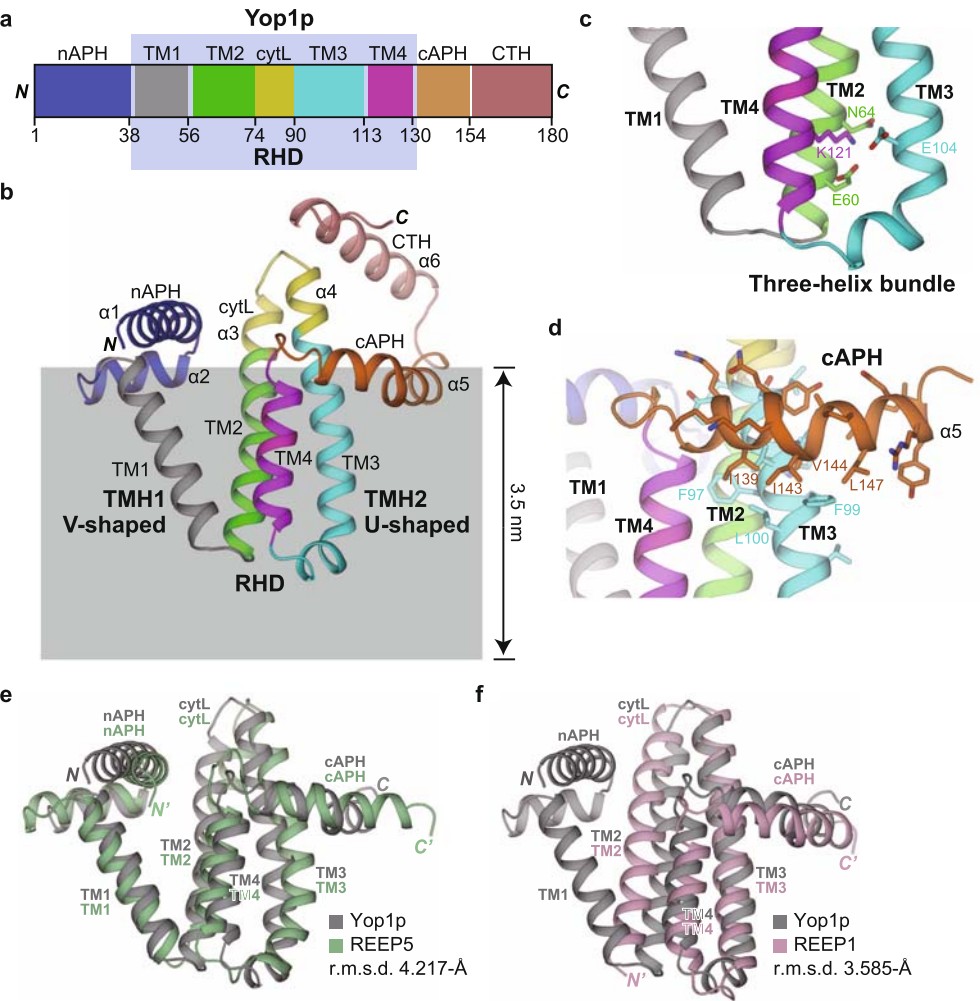

**Fig. 1 | Structures of REEP proteins. a** Schematic diagram of Yop1p. Domains are labeled and colored differently. Residue numbers for domain boundaries are shown at the bottom. The reticulon homology domain (RHD) is highlighted in the light blue box. APH, amphipathic helix; TM, transmembrane; cytL, cytosolic loop; CTH, C-terminal helix. **b** Structure of Yop1p predicted by RoseTTAFold. Regions of Yop1p are shown in cartoon representation and colored as in (**a**). The two transmembrane hairpins (TMHs) and the RHD are highlighted. Cytosolic helices are numbered. The estimated position of the lipid bilayer is shown as a gray box with its thickness labeled and the top aligned with the starts/ends of the TMs. **c** Structural details of a hydrophilic pocket within the TM regions. TM regions are colored as in (**b**) and side chains of key residues are shown as sticks. The pocket helps to hold a three-helix bundle. **d** Structural details of the C-terminal APH (cAPH). Domains are colored as in (**b**) and side chains of key residues are shown as sticks. Residues in the cAPH exhibit a clear amphipathic pattern, with hydrophobic residues pointing towards the membrane and hydrophilic residues pointing in the opposite direction. The hydrophobic face of the cAPH is laterally embraced mainly by residues from TM3. **e** Structural comparison of Yop1p and REEP5. Yop1p is shown in gray and REEP5 in pale green. Major domains are labeled. The root mean squared deviation (RMSD) measurements are indicated. **f** As in (**e**), but between Yop1p and REEP1. Yop1p is shown in gray and REEP1 in light pink.

and weakness of the lower limbs due to a length-dependent abnormality of the corticospinal axons. However, how ER tubule-forming proteins generate membrane curvature, and how their mutations cause HSP, is largely unclear. It is speculated that the RHD serves as a "wedge" insertion, occupying more space in the outer leaflet of the bilayer than the inner leaflet, inducing local curvature[3]. In addition, a conserved amphipathic helix (APH) has been found C-terminal to the RHD, which is necessary for functions of tubule-forming proteins, likely contributing to curvature generation in a similar manner[20,21]. RTNs and REEPs are known to form dimers and oligomers[6,7], which likely promote membrane bending as a "scaffold"[3]. How wedge and scaffolding each contributes to curve membranes is not known. Whether oligomerization directly links to curvature generation and, if so, how it is achieved remains elusive. Some of these proteins have been successfully purified[7,10,21,22], but no structural information is available thus far, likely due to the heterogeneity in the oligomeric states of the recombinant fragments and a lack of crystal packing surface because of the expected shape and configuration. Here, we

predict the structure of REEPs and RTNs using recently developed deep learning methods[23,24]. We confirmed the existence of TM hairpins and APHs and, thus, the possible wedge insertion. However, disruption of the wedged structures did not significantly prevent tubule generation in cells. Instead, TM-mediated dimerization and subsequent oligomerization, facilitated by the cytosolic loop (cytL) and C-terminal helix (CTH), modulate curvature generation. These results reveal the basis and importance of previously unappreciated assembly modes of the curvature-stabilizing proteins.

## Results

### Structural prediction of ER tubule-forming proteins

To investigate the mechanism of curvature generation by ER tubule-forming proteins, we first predicted the structure of Yop1p using RoseTTAFold[23]. As expected, Yop1p is mostly α-helical (Fig. 1b). The RHD region is comprised of two TM hairpins connected by the cytL (residues 75–89). The loop contains a helical extension from both TM2 and 3 (α3 and α4). The end of TM2 and the start of TM3 were first

determined by the appearance or disappearance of polar residues in the sequence, and then verified by cross-comparison with ends of other TMs to make sure that all TMs start or end at the same altitude. Within the membrane, the TM1/2 pair adopts a "V" shape with a bend at G56 and G57, whereas the TM3/4 pair forms a "U" shape with a turning point at P115. In addition, TM2-4 creates a three-helix bundle with mostly hydrophobic interactions but an unconventional hydrophilic pocket deep in the membrane. K121 in TM4 points at E104 in TM3 and E60 in TM2, stabilizing the bottom part of the bundle (Fig. 1c). Assuming a 3.5 nm thickness of the lipid bilayer, the TMs of Yop1p would less likely span the entire membrane (Fig. 1b).

The RHD is immediately followed by a previously identified APH (residues 134–152, α5, cAPH). It has a slight bend at V144, wrapping around the nearby TM3 (Fig. 1d). Interactions between these two helices ensure a locked position of the APH relative to membranes. Similarly, the N-terminus (NT) that precedes the RHD exhibits an APH-like pattern (residues 1–32, nAPH) (Supplementary Fig. 1a). Two helices (α1 and α2) that are almost perpendicular to each other embrace the beginning of TM1 and project their hydrophobic side towards the lipid bilayer where the TMs reside. Notably, both APHs lie on a plane that aligns with the cytosolic ends of all TMs (Fig. 1b). These observations suggest that, even though the prediction algorithm does not take the membrane into consideration, the configuration of Yop1p fits well with it being in lipids, likely reinforced by specific intra-molecular interactions.

We noticed that the region after cAPH, the C-terminus (CT), forms another helix (residues 158–180, α6, CTH) (Supplementary Fig. 1b). Because a loop exists between cAPH and CTH, the position of the CTH becomes flexible, varying among predicted models. Other than the CTH, prediction of the structure of Yop1p by AlphaFold is virtually identical to that of RoseTTAFold[24] (Supplementary Fig. 1c), strengthening the reliability of these results. Next, we predicted REEP5, the mammalian REEP closest to Yop1p. Consistent with their sequence similarity, the two structures can be superimposed with each other, except the CTH (Fig. 1e). We also analyzed the structure of REEP1, a representative of REEP1-4. REEP1 has a truncated NT and an extended CT compared to REEP5/6 and Yop1p. REEP1 has been speculated to possess a proportionally shortened TM1/2 hairpin[12]. However, the predicted structure of REEP1 contains three, not four, TMs, leaving out the TM1 seen in REEP5 and Yop1p (Fig. 1f). To test the conformation of the first hydrophobic segment of REEP1, we performed topology analysis using cysteine modification. The NT of REEP1 was inaccessible to cytosolic cys-modifier, but a site after the first hydrophobic segment was readily modified (Supplementary Fig. 1d). These results confirm that REEP1 lacks the entire TM1 seen in other RHDs but maintains the bundling of the remaining TMs.

Next, we compared the structures of REEPs to those of RTNs. The RHD of RTN4 adopts a similar organization as Yop1p (Supplementary Fig. 1e). Both TM1/2 and TM3/4 hairpins are V-shaped, with much larger angles than in Yop1p. The turning points of the two hairpins are in close proximity but no bundling of the TMs is observed. A similar cAPH is found after TM4, but no apparent anchoring of the helix is seen. The cytL of RTN4 is much longer than that of the REEPs and forms a helix-turn-helix. These results suggest that REEPs and RTNs utilize similar elements to shape membranes, but the relative position of these components is not conserved.

## TM-mediated assembly and curvature generation of Yop1p

ER tubule-forming proteins undergo homotypic and heterotypic oligomerization[5,6]. To probe interfaces for self-association, we introduced cysteine into Yop1p, which is naturally Cys-free, and performed chemical cross-linking. First, we chose residues in TMs that are outward facing in the predicted structure and possibly at similar altitude in the membrane. In this case, copper-o-phenanthroline (Cu:Phe) was used to catalyze disulfide bond formation for its good membrane permeability. Previous unnatural amino acid-based cross-linking has suggested that F68 in Yop1p TM2 (equivalent to F65 in *Schizosaccharomyces japonicus* Yop1) is very close to the pairing protomer in a Yop1 dimer[21]. We found that F68C formed a homotypic disulfide bond in Yop1p (Fig. 2a). Consistently, nearby residues, including F65 in TM2 and W94 in TM3, mediated similar dimer formation when mutated to Cys (Fig. 2b). These results suggest that TM2 of Yop1p consists of a homotypic dimer interface.

Next, we tested the other three TMs using the same strategy. TM4 exhibited a strong tendency for dimer formation, with V123C, F124C, and I126C forming significant disulfide bond-based dimers (Fig. 2c, d). In contrast, L125C, which faces inward in the TM helix bundle, did not mediate detectable dimer formation. Residues tested in TM3 (F97C, F99C, and L100C) mostly showed marginal dimer formation (Supplementary Fig. 2a, b). The approach of a neighboring TM3 is likely hindered by the existence of cAPH. Finally, some residues in TM1 (L43C and F45C), but not others (L49C), participated in self-association (Supplementary Fig. 2c, d). Given that three groups of disulfide bond-competent Cys mutants point in different direction, these results indicate at least three independent dimer interfaces in the TMs of Yop1p.

We also tested whether cytosolic regions come into proximity in the Yop1p assembly. When Cys was introduced into the NT, S2C displayed no apparent disulfide bond formation upon oxidation by diamide (Supplementary Fig. 2e). In contrast, D87C in the cytL had evident dimer formation (Supplementary Fig. 2e), consistent with its close association with the TM2 dimer interface. Unexpectedly, both S157C and S178C in the CTH formed a homotypic dimer (Supplementary Fig. 2e). The CTH dimer is likely a coiled coil (Supplementary Fig. 1b). Given the flexible linker preceding the CTH, the dimer is likely independent of the RHD.

Collectively, we identified multiple dimer interfaces of Yop1p. To further explore the oligomeric assembly of Yop1p, we focused on the center piece of the molecule, the RHD. We reasoned that the TM2 and TM4 interfaces could be combined for oligomerization. To this end, we 3D-printed the space-filling model of the core region of Yop1p (residues 1–154). When two copies of the module were assembled with either TM2 or TM4 facing each other, both dimers could be locked into a stable state (Fig. 2e, f). The dimers could be extended into tetramers by alternating the two interfaces (Fig. 2g, h). Strikingly, both tetramers had a curved shape, coinciding with the curvature seen in a tubular membrane of ~30 nm diameter (Fig. 2i). The assembly of 3D-printed models indicates high confidence in the structural prediction and that membrane curvature in ER tubules is mainly generated and stabilized by curved Yop1p oligomers.

In the manually assembled TM4-based dimer, we noticed that a pair of bulky residues makes extensive contact in the bottom of the TMs (Fig. 2f, bottom view). We identified the site to be F116 in the beginning of the TM4. As predicted, when F116 was mutated to Cys, it formed strong homotypic cross-linking upon oxidation (Supplementary Fig. 2f), whereas a similar mutant of a nearby hydrophobic residue, W118C, exhibited no apparent disulfide bond formation (Supplementary Fig. 2f). We also tested the co-existence of various dimer interfaces by introducing two Cys substitutions in Yop1p-HA. The three double mutants (F65C/V123C, TM2/TM4; F65C/S178C, TM2/CTH; V123C/S178C, TM4/CTH) all had higher order oligomer formation (Supplementary Fig. 3a). By contrast, the double mutants F65C/F68C or F116C/V123C formed only dimers (Supplementary Fig. 3b), consistent with the two being in the same interface. Taken together, these data confirm that the manual assembly indeed reveals critical interfaces and a resulting mode of oligomerization.

## Systematic structure-function analysis of Yop1p

Our oligomerization analysis revealed the importance of Yop1p-TMs in curvature generation. Previous studies have emphasized the role of

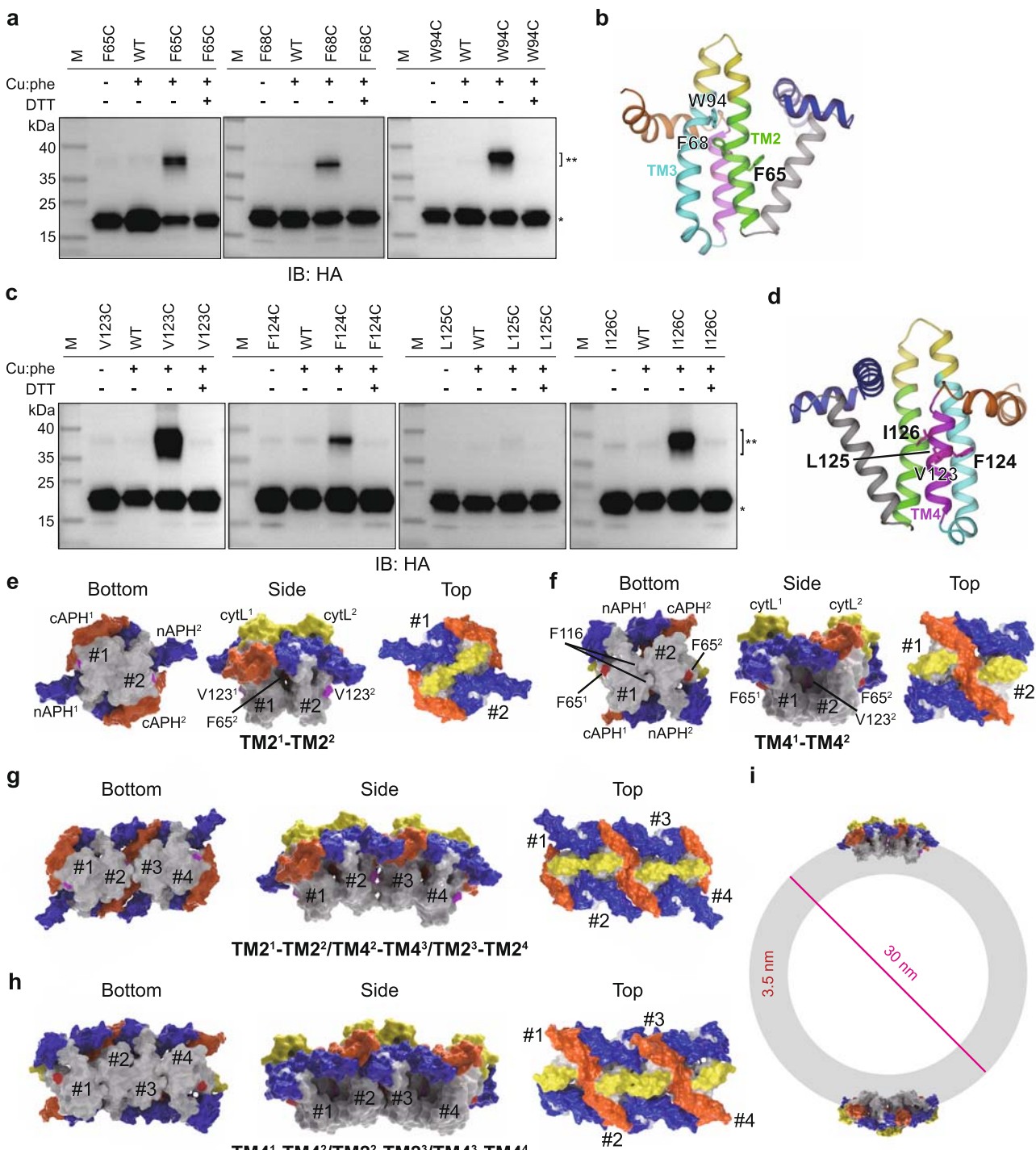

**Fig. 2 | Assembly of Yop1p. a** Yop1p dimerization probed by Cys-cross-linking. Membrane fractions of yeast cells expressing HA-tagged Yop1p, including wild-type (WT) or Cys mutants close to TM2, were treated with 1 mM copper-o-phenanthroline (Cu:Phe). Reduction by 100 mM DTT was used as a control. Samples were separated by non-reducing SDS-PAGE and immunoblotted (IB) by anti-HA antibodies. A single asterisk (*) indicates the monomer and double asterisks (**) the dimer. M, molecular marker. **b** Positions of Cys-replaced residues used in (**a**). **c** As in (**a**), but with residues in TM4. **d** Positions of Cys-replaced residues used in (**c**). **e** TM2-mediated dimerization of Yop1p. The space-filling model of Yop1p (residues 1–154) is 3D-printed, manually assembled, scanned, and rendered by Maya software. The cytosolic regions of Yop1p are colored as in Fig. 1b and the TM regions are colored white. F65 in TM2 is highlighted in red and V123 in TM4 in magenta. Top,

side, and bottom views are shown with molecules numbered. The dimer was assembled with the TM2s facing each other. TM transmembrane, cytL cytosolic loop, nAPH N-terminal amphipathic helix, cAPH, C-terminal amphipathic helix. **f** As in (**e**), but with TM4-mediated dimerization. **g** As in (**e**), but with TM4-centered tetramerization. The tetramer is built with a TM4-based dimer in the middle, flanked by two TM2-based assemblies on each side. **h** As in (**e**), but with TM2-centered tetramerization. The tetramer is built with a TM2-based dimer in the middle, flanked by two TM4-based assemblies on each side. **i** The two tetramers of Yop1p are fitted into a cross-section of a tubule with 30 nm diameter. The membrane is shown as a gray ring with its thickness set to 3.5 nm. Yop1p models are placed proportionally. Source data are provided as a Source data file.

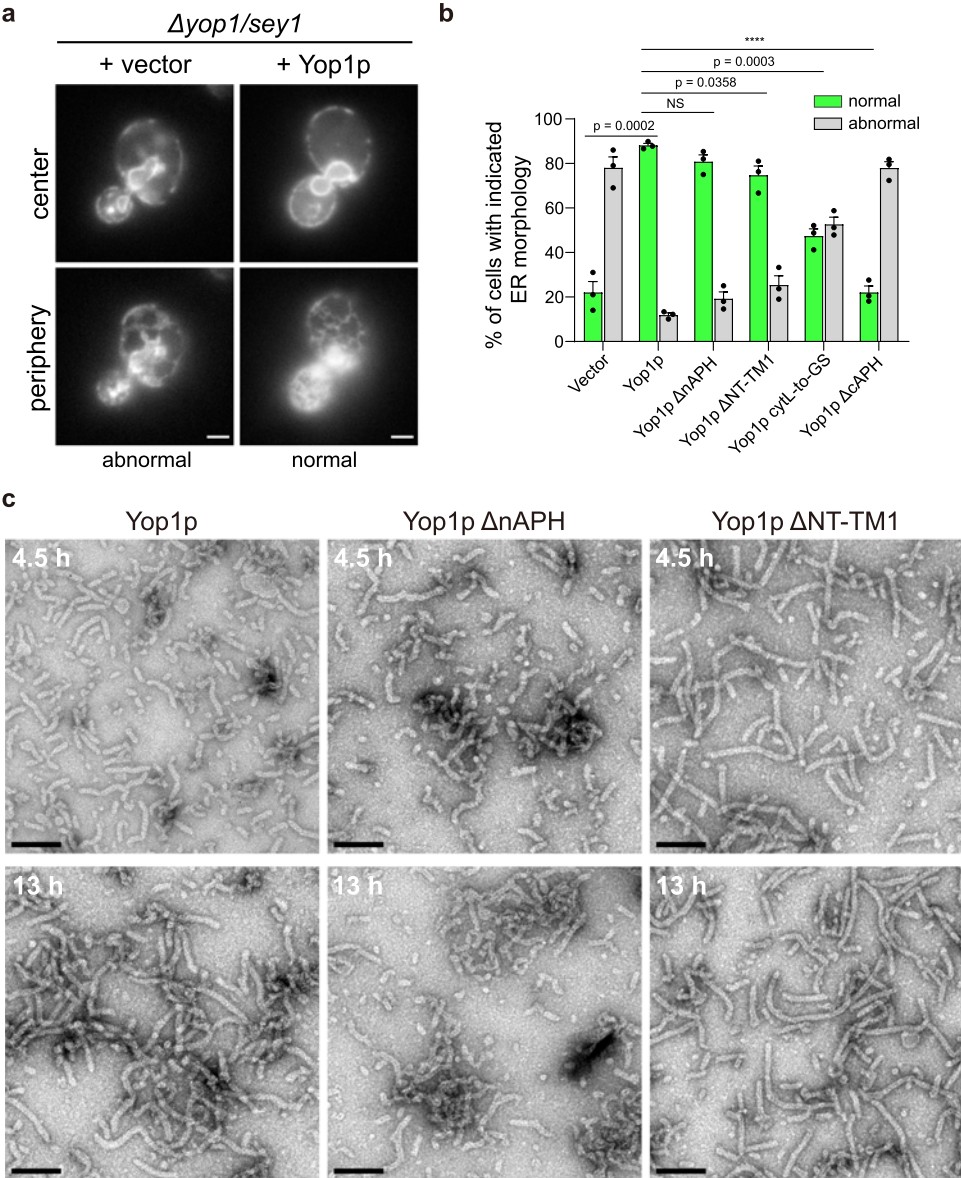

**Fig. 3 | Domain analysis of Yop1p. a** ER morphology visualized in yeast cells. A GFP fusion of the ER protein Sec63p (Sec63p-GFP) was expressed in cells lacking Yop1p and Sey1p (*Δyop1/sey1*). Vector or Yop1p-expressing construct was transformed into the cells. The localization of the protein, indicative of ER morphology, was determined by fluorescence microscopy, with the microscope focused either at the center or the periphery of the cell. Categories of ER morphology were defined as "normal" or "abnormal". Scale bars, 2 μm. **b** Quantification of ER morphology. Vector or indicated Yop1p constructs were transformed into *Δyop1/sey1* cells. The ER was visualized as in (**a**) and categorized by counting at least 100 cells per sample. The quantitative data are shown as mean ± SEM of three repeats. NS, not significant; ****$p < 0.00001$, unpaired two-sided Student's t-test. **c** In vitro tubule formation by Yop1p. Yop1p, wild-type or mutant, was mixed with *E. coli* polar lipids in LDAO, and the detergent was removed with Bio-beads over 4.5 or 13 h. The resulting proteoliposomes were analyzed by negative-stain EM. Scale bars, 100 nm. Source data are provided as a Source data file.

cAPH in ER tubule formation[20]. Therefore, we set out to investigate the contribution by other regions of Yop1p. When Yop1p and Sey1p, a GTPase that mediates ER fusion, were deleted in yeast cells, the morphology of the ER was compromised, with significant loss of the tubular network[25]. Reintroduction of wild-type Yop1p at endogenous levels efficiently restored the ER structure (Fig. 3a, b and Supplementary Fig. 4a). When the NT, which is mostly nAPH, or NT-TM1 was deleted in Yop1p, the truncated proteins were able to maintain the ER morphology (Fig. 3b), replacing the wild-type protein. In contrast, cAPH deletion was largely defective (Fig. 3b), as described previously[20,21]. These results suggest that any wedge insertion by nAPH and the wedge-shaped TM hairpin play a minor role in tubule formation by Yop1p. Notably, the ΔNT-TM1 mutant of Yop1p resembles the domain architect of REEP1 (Fig. 1f), implying that shaping proteins with

three TMs, which lose the TM wedge, should still be able to stabilize the membrane curvature.

To confirm that the TM wedge plays a less important role in curvature generation, we performed assembly of 3D-printed models of Yop1p ΔNT-TM1. When two copies of the module were assembled with either TM2 or TM4 facing each other, dimers could be assembled, but are less stable than that of the wild type (Supplementary Fig. 3c, d). The dimers could still be extended into tetramers by alternating the two interfaces (Supplementary Fig. 3e, f), and both tetramers remained a curved shape, coinciding with the curvature seen in a tubular membrane of ~30 nm diameter (Supplementary Fig. 3g). These results confirm that the three-TM tubule-forming protein could still generate membrane curvature once they assembled into oligomeric scaffold.

Next, we tested the involvement of the cytL and CTH. When residues in the cytL were all substituted with alanine, mutated Yop1p (cytL-to-A) was poorly expressed (Supplementary Fig. 4b). When the loop was replaced by a GS-based linker (GSGSS), the mutant (cytL-to-GS) was expressed similar to the wild-type but with a ~50% reduced ability for maintaining ER morphology (Fig. 3b). Deletion of the CTH resulted in very little expression in cells (Supplementary Fig. 4b). These results suggest that the cytL and CTH are critical for the stability of the protein and that at least the cytL is involved in the function of Yop1p.

To further confirm the function of individual Yop1p segments, we purified mutated Yop1p and performed in vitro tubule formation assays (Supplementary Fig. 4c, d). Yop1p was incubated with detergent-solubilized *E. coli* polar lipids. Reconstitution of proteoliposome was achieved by detergent removal with bio-beads. As expected, reconstituted wild-type Yop1p shaped the membrane into tubular structures (Fig. 3c), which were observed by negative staining transmission electron microscopy (TEM). When ΔNT or ΔNT-TM1 was purified and reconstituted, similar tubular membranes were seen (Fig. 3c). The tubules generated by ΔNT-TM1 were longer and straighter compared to those generated by the wild-type (Fig. 3c). These results confirm that the nAPH and TM1 of Yop1p are dispensable for tubule formation.

We also purified the cytL-to-GS mutant and reconstituted it into proteoliposomes (Supplementary Fig. 4c, d). Within 4.5 h of detergent removal, short tubules were seen by TEM (Supplementary Fig. 4e). After extensive reconstitution (13 h), longer tubules could be detected, but not as frequent as with wild-type, and they were relatively straight like the tubules obtained with ΔNT-TM1 (Supplementary Fig. 4e). These data suggest that the cytL affects the efficiency of tubule formation, but the in vitro assay is generally less sensitive than the in-cell rescue assay.

## Comparison of Yop1p and REEP1/REEP5

Next, we tested whether the mechanism of Yop1-mediated tubule formation is conserved among the REEP family (Fig. 4a). When REEP1 and REEP5 were expressed in *Δyop1/sey1* cells to the equivalent levels as Yop1p, they failed to restore the ER morphology (Fig. 4b and Supplementary Fig. 5a). The same results were obtained when the REEPs were reintroduced into *Δrtn1/rtn2/yop1* cells[5] (Supplementary Fig. 5b), which have similar ER morphology defects as the *Δyop1/sey1* cells but different genetic compositions. Consistent with previous reports[21], purified human REEP1 or REEP5 could only form lipoprotein particles (LPP), not tubules, when incorporated into lipids in vitro (Fig. 4c and Supplementary Fig. 5c, d). These surprising results suggest that, even though the three proteins have high sequence similarity, the tubule-forming capacity of the mammalian REEPs is weaker than that of Yop1p.

We reasoned that subtle differences in amino acid composition account for the differentiated tubule-forming capacity of Yop1p, REEP1, and REEP5. Therefore, sequences in key oligomerization sites were aligned (Supplementary Fig. 5e). In TM2 and TM4, where dimer formation was detected, we found that Yop1p possesses bulky hydrophobic residues compared to REEP1 and REEP5. Therefore, we performed residue swapping between Yop1p and REEP5. Although Yop1p F65A/V123A exhibited normal ER tubule formation ability, REEP5 L61F/G119V had an enhanced rescue capability (Fig. 4b). Next, we analyzed the overall oligomerization using a previously established ethylene glycol bis(succinimidylsuccinate) (EGS) cross-linking assay[7]. REEP5 L61F/G119V exhibited a slight increase in oligomerization compared to wild-type (Supplementary Fig. 5f). These results suggest that the strength of TM-mediated dimerization partially explained the difference between Yop1p and REEPs.

We also investigated whether the cytL and CTH are different between Yop1p and REEPs. To this end, we replaced the cytL or CTH of REEP with the cytL or CTH of Yop1p (Supplementary Fig. 5a). The chimeric protein REEP5 cytL^Y exhibited a slight increase in rescued defects in the ER morphology compared to wild-type REEP5 (Fig. 4b). REEP5 1/2CTH^Y (with first half of the Yop1p CTH implanted) yielded an ~50% increase (Fig. 4b). Strikingly, REEP1 1/2CTH^Y exhibited nearly full capacity in replacing Yop1p (Fig. 4b). These results suggest that the strength of cytL and CTH-mediated dimerization was also involved in the difference between Yop1p and REEPs. We then performed an in vitro tubule formation assay using REEP-Yop1p chimeric proteins (Supplementary Fig. 5c). Though REEP5 1/2CTH^Y failed to generate a tubular structure upon reconstitution, both REEP1 1/2CTH^Y and REEP1 CTH^Y efficiently caused membrane tubulation in vitro (Fig. 4c). These results confirm that the tendency for CTH dimerization regulates the capacity of ER tubule formation by the REEP family of proteins.

Mutations in human REEP1, termed SPG31, are linked to HSP[12,26,27]. Several point mutations have been found in the RHD region of REEP1[27-31] (Fig. 4d). P19R and A20E localize to the end of TM2 (the first TM of REEP1) and are likely involved in cytL stability. W42R in TM3 is equivalent to W94 in Yop1p, which participates in TM2-mediated dimerization. I70R in TM4 resembles T122 in Yop1p, which regulates TM4-mediated dimerization. To test whether these mutations cause HSP by compromising oligomerization and ER tubule formation, we introduced these changes into REEP1 1/2CTH^Y and expressed them in *Δyop1/sey1* cells. All mutants had defects in maintaining ER tubule formation, with I70R being the most severe (Fig. 4e). We subsequently performed the EGS assay and noted reduced oligomerization of the I70R mutant (Supplementary Fig. 5g). Similarly, it was much less efficient in forming tubules when purified and reconstituted (Fig. 4c). We confirmed that the defective tubule formation by REEP1 CTH^Y I70R was less likely due to the choice of lipids during reconstitution; when yeast polar lipids were used, REEP1 CTH^Y formed tubule properly but the I70R mutant failed (Supplementary Fig. 4f). Collectively, HSP-causing mutations can be directly associated with the regulation of dimerization and oligomerization, confirming the importance of REEP assembly.

## Discussion

We previously showed that RTNs and REEPs are sufficient for tubule formation and form oligomers[7]. With the help of AI-based structural prediction, we systematically analyzed the molecular basis of REEP oligomerization and the mechanism of curvature generation by these proteins. We confirmed that TM2 is involved in dimer formation, as previously reported[21], and revealed the homotypic nature of the association using Cys-based cross-linking. Furthermore, we found TM4, TM1, cytL, and CTH as previously unidentified, additional dimer interfaces. TM4, yet another dimer interface on the back of the TM2 dimer site, would convincingly explain how biochemically observed oligomers are generated by combining different dimers. The TM2 and TM4-mediated dimer formation constitutes the core driver of Yop1p RHD assembly. The cytLs likely come close to each other in the setting of the TM2 dimer (Fig. 2e). Similarly, the nAPHs pack against each other in that dimer (Fig. 2e), whereas the cAPHs stay side-by-side in the TM4 dimer (Fig. 2f). The CTH possibly dimerizes independently. These interfaces, even though individually weak, were systematically verified by cross-linking, in cell rescue experiments, and in vitro tubule formation assays. Collectively, a combination of various dimeric interaction modes leads to substantial oligomerization. The multifaceted self-assembly of Yop1p is critical for curvature generation and ER tubule formation, because altered dimer interfaces cause compromised tubular ER shaping and weakened dimer interfaces, as seen in REEP1 and REEP5, in correlation with reduced ER tubule formation.

Importantly, alternating self-association using TM2 and TM4 interfaces generates a curved oligomer with an overall bending capacity equivalent to the curved membranes seen in an ER tubule (Fig. 2i). Our results suggest that an oligomeric "scaffold" plays a predominant role in curvature generation. The previously believed "wedge" insertion, even though still applicable to cAPH, appears to be less

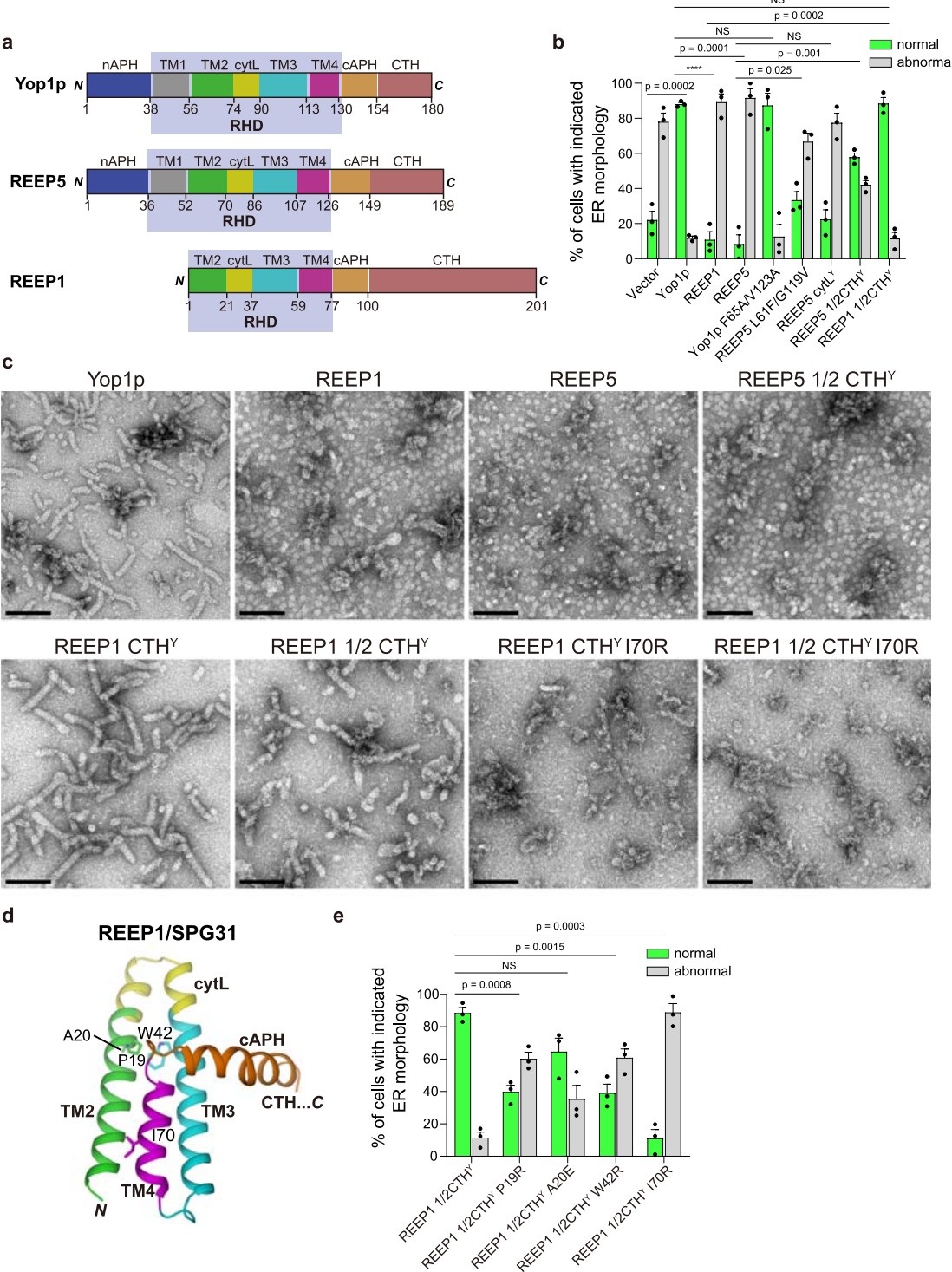

**Fig. 4 | Comparison of REEP proteins. a** Schematic diagram of REEP proteins. Domains are labeled and colored differently. **b** ER morphology rescued by the REEP proteins. Sec63p-GFP was expressed in cells lacking Yop1p and Sey1p (*Δyop1/sey1*). Vector or indicated REEP constructs were also transformed into these cells. The ER was visualized and categorized by counting at least 100 cells per sample. The quantitative data are shown as mean ± SEM of three repeats. NS, not significant; ****$p < 0.00001$, unpaired two-sided Student's t-test. **c** In vitro tubule formation by REEP proteins. REEP, wild-type or chimeric mutant, was mixed with *E. coli* polar lipids in DDM, and the detergent was removed with Bio-beads for 4.5 h. The resulting proteoliposomes were analyzed by negative-stain EM. Scale bars, 100 nm. **d** HSP mutations in RHD of REEP1. REEP1 structure is shown in cartoon representation with major domains labeled as in (**a**) and select SPG31 mutations highlighted as sticks. (**e**) As in (**b**), but with REEP1 1/2CTH$^Y$, wild-type or SPG31 mutants. NS, not significant, unpaired two-sided Student's t-test. Source data are provided as a Source data file.

important, as demonstrated by the NT deletions in Yop1p, which eliminated the potential nAPH and disrupted the V-shaped TM hairpin. Consistently, the truncated mutant was able to form curved oligomers just like wild-type Yop1p. Similarly, REEP1 and REEP5, possessing a perfect cAPH for wedge mechanism, and yet fail to form tubules in vitro and replace Yop1p in cells, due to reduced oligomeric scaffolding. In the assembled 3D-printed model, the APHs frequently pack side-by-side, suggesting that they might have extra roles in mediating

self-association. Both APHs also facilitate the positioning of their connecting TMs, which indirectly involve in oligomer formation. The presence of these APHs ensures that TM2 and 4, but not 1 and 3, are the main front of self-association.

Our findings also establish the requirement of oligomer formation in generating ER tubules. REEP1- and REEP5-formed LPP was previously proposed as examples of extreme curvature, in which splayed dimers were formed[21]. We showed that REEP1 and REEP5 in fact had weaker ability in generating ER tubules than that of Yop1p, and modified REEP1 with enhanced self-association of CTH were able to form tubules, but not LPPs, in vitro and replace Yop1p in cells. Taken together, our model emphasizes the formation of oligomeric, but not dimeric, scaffold in stabilization membrane curvature.

Our results reveal that mammalian REEPs are weaker tubule-forming proteins than the yeast REEP. It is possible that, by having redundant genes in the RTN and REEP families, the individual mammalian REEPs could evolve into less potent shaping proteins. Alternatively, yeast cells and mammalian cells demand differentiated ER tubule-forming capacity. The mechanism of multifaceted assembly ensures a flexible and fine-tuned tubular ER network. The same concept is probably used by the RTN and other RHD-containing proteins and may be a common theme for organelle shaping by integral membrane proteins.

## Methods
### Structural analysis
Structural prediction was performed with RoseTTAFold and Alpha-Fold. Comparisons and illustrations were prepared using the program PyMOL (https://pymol.org). Models of the predicted Yop1p molecule with a surface representation were printed by 3D printer (Lite600, UnionTech) with opaque white, low-viscosity liquid photopolymer (SOMOS Imagine 8000, Royal DSM). Models were then manually assembled and glued for stabilization, scanned by KSCAN20 3D scanner (ScanTech), and rendered and colored using Maya 3D visual effects software (Linghou Tech., Wuhan, China).

### *S. cerevisiae* strains and constructs
The following yeast strains were used: BY4741 (*MATa his3Δ1 leu2Δ2 met15Δ ura3Δ*), JHY4 (BY4741 *sey1Δ::kanMX4 yop1Δ::HIS3MX6*), NDY257 (BY4741 *rtn1Δ::kanMX4 yop1Δ::kanMX4 rtn2Δ::kanMX4*), and FM135a (*MATa ura3-52, leu2-3.112, reg1-50, gal1, pep4-3, prb1-112*). For expression of Yop1p at endogenous levels, the full coding region of Yop1p plus a C-terminal HA tag and 300 bp upstream sequences was amplified and inserted into the BamHI/XhoI site of pYC2/CT (a URA3/CEN plasmid) with the original GAL promoter removed[25]. Similarly, the nAPH and cAPH deletions of Yop1p plus a C-terminal HA tag with their endogenous promoters were inserted into the pYC2/CT vector, whereas the TM1 deletion of Yop1p was cloned into the BamHI/XhoI site of pESC-URA (a 2μ plasmid) with the endogenous promoter and the original GAL promoter removed. All Yop1p cysteine variants were made from the Yop1p-HA construct in pYC2/CT using the QuikChange Site-Directed Mutagenesis Kit (Stratagene).

The codon-optimized *H. sapiens* REEP1 and REEP5 genes plus a C-terminal HA tag and Yop1p endogenous 500 bp upstream and downstream sequences were amplified and inserted into the BamHI/XhoI site of pESC-URA (a 2μ plasmid) with the original GAL promoter removed. The TM2/4 residues or cytL or CTH of REEPs were replaced with the individual Yop1p segments using the QuikChange Site-Directed Mutagenesis Kit (Stratagene). Four SPG point mutations in the RHD region of REEP1 were introduced into REEP1 1/2CTH[Y] in the same way. All constructs were verified by DNA sequencing (AZENTA).

### Fluorescent microscopy
The plasmid pWP1098 (a LEU2/CEN plasmid) expressing Sec63p-GFP was used to visualize ER morphology as described previously[7]. Yeast cells were imaged live in growth medium using the DeltaVision OMX V3 imaging system (Cytiva, GE Healthcare), a ×100/1.40 NA oil objective (Olympus UPlanSApo), solid-state multimode laser 488, and an electron-multiplying charge-coupled device (CCD) camera (Evolve 512×512, Photometrics).

### Membrane topology analysis
All REEP1 cysteine mutants were created from REEP1-HA construct using the QuikChange Site-Directed Mutagenesis Kit (Stratagene). Maleimide PEG 5 kDa (MP) modification experiments were performed as described previously[5]. Briefly, COS-7 cells transfected with REEP1 constructs were washed in HCN buffer (50 mM Hepes [pH 7.5], 150 mM NaCl, 2 mM CaCl₂) and then treated with 0.005% digitonin or 1% Triton X-100 for 20 min at 4 °C. Samples were incubated for 30 min at 4 °C with 1 mM MP in the absence or presence of 20 mM DTT. DTT and 1% Triton X-100 were then added to all samples. Following SDS-PAGE, immunoblotting was performed with anti-HA antibody (CST, 3724S).

### Cysteine cross-linking
BY4741 yeast cells were transformed with individual Yop1p cysteine mutants. The cells were cultured at 30 °C in synthetic medium (-URA) to an OD$_{600}$ of approximately 1.0, harvested by centrifugation, and washed with PBS. Yeast microsomal membrane fractions were prepared as described previously[32]. Briefly, cells were pelleted and converted to spheroplasts by treated with 2 mL of 2% snailase in sorbitol buffer (1 M sorbitol, 0.02 M sodium citrate, 0.1 M EDTA, 0.02 M Na₂HPO4, [pH 5.8]) at 30 °C for 1.5 h, washed twice with 2 mL of sorbitol buffer on ice, swollen in 1 mL of lysis buffer (800 mM sorbitol, 10 mM triethanolamine, 1 mM EDTA [pH 8.0], and protease inhibitor cocktail) for 15 min on ice, and homogenized with 35 strokes in a tight-fitted Dounce homogenizer. Crude homogenates were centrifuged at $1000 \times g$ (rotor FA-45-24-11, Eppendorf) at 4 °C for 5 min to remove the nucleus and unbroken cells. The supernatant was further centrifuged at $20,000 \times g$ (rotor FA-45-24-11, Eppendorf) at 4 °C for 30 min to remove heavy organelle fractions. The resulting supernatant was centrifuged at $100,000 \times g$ (rotor TLA 100.3, Beckman) at 4 °C for 1 h. Then the resuspended membrane fractions were treated with 1 mM copper-o-phenanthroline (Cu:Phe) or 2 mM diamide for 30 min at room temperature, and the reactions were stopped by the addition of 10 mM N-ethylmaleimide for 15 min at 4 °C. For reduced controls, 100 mM DTT was added. Following SDS-PAGE, immunoblotting was performed with anti-HA antibody (CST, 3724S).

### EGS cross-linking
Ethylene glycol bis(succinimidylsuccinate) (EGS, Pierce) was dissolved in anhydrous dimethyl sulfoxide (DMSO) and diluted to the desired concentration. Cross-linking was performed as described previously[6]. Briefly, the membrane pellet, as obtained in cys-cross-linking, was resuspended in HKM buffer (25 mM Hepes [pH 7.8], 150 mM KCl, 2.5 mM MgCl₂) to a final volume of 80 μl, and then 1 μl EGS was added into each 20 μl membrane aliquot for 30 min at room temperature. The reactions were quenched with 2 μl of 1 M Tris [pH 7.5] for 15 min. Samples were separated by a 4–20% SDS-PAGE and immunoblotting performed with anti-HA antibody (CST, 3724S).

### Protein expression and purification
Yop1p was expressed and purified as described previously[7]. Briefly, FM135a strain was transformed with His-Yop1p-HA expression constructs, grown in SC-URA with 2% glucose to an OD$_{600}$ of approximately 1.0, and induced by 2% galactose for 10 h. The cells were resuspended in TKG buffer (50 mM Tris [pH 7.5], 150 mM KCl, 5% glycerol) and disrupted by low temperature ultra-high pressure cell disrupter (JN-Bio). The lysate was ultra-centrifuged at $119,000 \times g$ for 1 h at 4 °C using a Type 45 Ti rotor (Beckman Coulter) to sediment the membranes. The membranes were solubilized in TKG + 1% LDAO

(Anatrace) for 1 h at 4 °C. Insoluble material was removed by centrifugation at 37,700 × *g* for 1 h. The supernatant was subjected to an Ni-NTA column (GE Healthcare) and bound material eluted with 0.5 M imidazole. The eluate was further purified by a HiTrap desalting column (GE Healthcare) in TKG + 0.05% LDAO. The His tag was removed by incubation with His-tagged thrombin for 1 h at room temperature. Thrombin and uncleaved proteins were isolated by Ni-NTA. Some mutated Yop1p constructs were tagged with C-terminal StrepII tag for affinity purification. For these proteins, solubilized membrane supernatants were mixed with Strep-Tactin XT 4Flow agarose resin (IBA) and incubated at 4 °C for 2 h with gentle agitation. The bound proteins were washed in 20 volumes of TKG buffer containing 0.05% LDAO and eluted with buffer containing 50 mM biotin.

The *Homo sapiens* REEP1 and REEP5 genes were cloned into pcDNA-4TO vector with a CMV promoter and C-terminal HA tag followed by a StrepII tag. All REEP-Yop1p chimeric proteins were created from REEP construct using the QuikChange Site-Directed Mutagenesis Kit (Stratagene). All wild-type and chimeric REEP proteins were overexpressed using Expi293F™ mammalian cells (Invitrogen), cultured in SMM 293T-II medium (Sino Biological, Inc.) and Expi293™ expression medium (Gibco) mixture in a 3:1 volume ratio, at 37 °C under 8% $CO_2$ in a shaker incubator at 120 rpm. When the cell density reached $2.0 \times 10^6$ cells/mL, the plasmid was transiently transfected into the cells. Three days post-transfection, cells were harvested, washed with cold PBS, and flash frozen at −80 °C. Cell pellets were resuspended in lysis buffer (25 mM Hepes [pH 7.4], 150 mM NaCl, 5% glycerol, 1 mM DTT, and protease inhibitors) and the membranes solubilized for 2 h with 1% DDM (Anatrace) at 4 °C. The clarified supernatant was incubated with Strep-Tactin XT 4Flow agarose resin (IBA) for 2 h at 4 °C. The resin was washed with 20 volumes of lysis buffer with 0.0174% DDM and eluted with 1× BXT buffer (100 mM Tris [pH 8.0], 150 mM NaCl, 1 mM EDTA, 50 mM biotin) containing 0.0174% DDM. All proteins were concentrated before reconstitution.

### In vitro tubule formation
The REEP proteins were reconstituted as described previously[7]. Briefly, detergent solubilized lipids and protein were mixed at the desired concentrations in a final volume of 100 μL. To remove the detergent, 5 mg of Bio-beads (Bio-Rad) were added every hour up to 4.5 h, and then continued to 13 h under stirring. All reconstitutions were carried out at 21 °C in a Thermo mixer at 750 rpm.

### Negative stain EM
Negative staining was done with 2% uranyl acetate and the procedures as described previously[7]. Briefly, a drop of 5 μL sample solution was spotted onto a glow-discharged carbon-coated copper grid (Zhongjingkeyi Tech, Beijing, China) for 20 s before blotting with filter paper. The grids were then washed with two drops of deionized water and stained with two drops of 2% uranyl acetate, blotting immediately after the first drop and incubating the second for 20 s. Excess uranyl acetate was blotted, and the grids were dried. The images were collected at room temperature using a Hitachi HT7700 transmission electron microscope at an acceleration voltage of 80 kV. Images were recorded at a magnification of 80,000 or 100,000. All images were recorded by a Gatan 832 CCD camera and processed with Image J software.

### Statistics and reproducibility
GraphPad Prism 8 software was used to perform statistical analyses. For ER morphology rescue assay, unpaired two-sided Student's t-test was used to determine significance between groups and numeric data are presented as mean ± SEM as indicated in the figure legends. All experiments were repeated three times, except that the in vitro tubule formation assay were repeated twice, and all attempts at replication with similar results.

### Reporting summary
Further information on research design is available in the Nature Portfolio Reporting Summary linked to this article.

## Data availability
All data generated or analyzed during this study are included in this article and its supplementary information files. Source data are provided with this paper.

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

## Acknowledgements

We thank Dr. Alicia Prater for proofreading, Xing Jia and Qing Bian for helping with the fluorescent imaging (Center for Biological Imaging, Institute of Biophysics, CAS), and Yanbao Tian for TEM imaging (Institute of Genetics and Developmental Biology, CAS). J.H. is supported by grants from the National Natural Science Foundation of China (32230024), the National Key R&D Program of China (2021YFA1300800), the Strategic Priority Research Program (XDB39000000), and Project for Young Scientists in Basic Research (YSBR-075) of the Chinese Academy of Sciences.

## Author contributions

J.H. designed and supervised the project. Y.X. performed most of the in vitro experiments. R.L. analyzed ER morphology in yeast cells. J.H. wrote the manuscript with input from all authors.

## Competing interests

The authors declare no competing interests.
