## [Peer Review File · Nature Communications]

Oligomeric scaffolding for curvature generation by ER tubule-forming proteinsREVIEWER COMMENTS

Reviewer #1 (Remarks to the Author):

This manuscript critically examines the mechanistic basis for how reticulons and REEPs induce, maintain, and stabilize the tubular morphology of the ER. The authors use clever strategies of exploiting structures of these proteins from AI-predicted algorithms and in the process validate these algorithms and establish a basic design principle on what constitutes fundamental determinants for these proteins to induce membrane curvature. The manuscript asserts that the contributions of amphipathic helices in these proteins is secondary to their ability to oligomerize and propagate their intrinsic curvatures across a large membrane surface. The results presented in the manuscript are arrived at from rigorous experiments and the description and inferences from primary data are quite valid. Furthermore, assessment of mechanism by which mutations found in hereditary spastic paraplegia better relate not to the intrinsic curvature-associated wedging features of these proteins but to their inability to self-associate into oligomers is quite a significant finding.

Oligomerisation among REEPs and reticulons have been previously reported to be an important determinant for their membrane curvature-inducing properties. The authors themselves cite these paper in their manuscript (Brady et al. (2015) PNAS, Wang et al. (2021) Nat. Commun.). An obvious consideration therefore is the novelty of the results. Based on the data presented here, this manuscript suggests that the prime determinants for curvature induction lies not in the ability for these proteins to wedge into one monolayer but from their ability to self-associate. This is quite a significant departure from current models. But the present manuscript just lays out these significant findings quite flatly without giving any opportunity for the reader to compare and contrast results presented here with those reported earlier. If the authors indeed believe that the major determinant for curvature induction among reticulons and REEPs is their ability to self-associate then this should be stated clearly and explicitly and would warrant a more fleshed out discussion section emphasising this novel finding.

The term 'multifaceted' in the title and text is quite vague and perhaps even misleading. If the authors assert that the dominant mechanism for curvature induction is the self-association among reticulons and REEPs then this should be stated clearly in the manuscript and reflected in the title and the text.

Experiments based on EGS crosslinking strategy suggest that REEPs and reticulons predominantly exist as dimers. Why would these proteins not form oligomers? This would admittedly be a prerequisite for them to propagate their intrinsic curvature across micron-long ER tubules.

A significant claim from the authors is that the predominant curvature-inducing determinant among REEPSs and reticulons is their ability to oligomerize and not so much that they wedge the membrane. Can this be substantiated by an analysis of the intrinsic curvature of the Yop1 and the deltaNT-TM1 Ypo1. This can be done by analysing the 3D printed models for their interfacial area to volume ratios. Perhaps the reason why deletion of the NT-TM1 in Ypo1 does not compromise its functions lies in the fact that delta NT-TM1 still retains an intrinsic curvature

.

Reconstitution of purified proteins into liposomes uses the E. coli polar lipid extract, which is strange choice of lipids. Could the formation of tubules be attributed to the choice this lipid mix. Were other lipid mixtures tested? The E. coli polar extract contains a higher proportion of PE, which by itself could facilitate the formation of curvature owing to its tendency to exist in the hexagonal phase. The manuscript would benefit from an analysis of reconstitution of proteins in a membrane with a lipid composition that otherwise prefers to remain in the lamellar phase. This could also test the true potential of reticulons to induce curvature.

Although reported earlier, it's curious that the aberrant ER morphology seen in Yop1/Sey1 deletion strains can be rescued with just the expression of Ypo1. Does this mean that membrane fusion by Sey1p is not necessary or that Sey1p functions redundantly with Ypo1 as a curvature inducing protein?

Thomas Pucadyil

Reviewer #2 (Remarks to the Author):

In this manuscript, the authors used RoseTTAFold, AlphaFold, and 3D-printed models to predict the structures of yeast Yop1p and related human REEP proteins. Structure prediction results and crosslinking analysis data together indicate that there exist two different homotypic interaction modes, with TM2 and TM4 forming the interaction interface, respectively. Curved-shaped tetramers can be formed by alternately using these two interaction modes. These findings provide novel insights into how these proteins generating membrane curvature and forms ER tubules.

I have the following comments and suggestions for the authors to consider.

Major points:

(1)

The Brady et al. 2015 PNAS paper proposed that Yop1p contains four hydrophobic helices long enough to span the ER bilayer fully (see Figure 8 of that paper).

In Figure 1B, the TMs of Yop1 are shown as not spanning the membrane bilayer.

The authors should explain why they choose to show the TMs this way and discuss why they do not agree with the model of Brady et al.

(2)

Line 93

"Notably, both APHs lie on a plane that aligns with the cytosolic ends of all TMs (Fig. 1B)."

Line 265

"P19R and A20E localize to the end of TM2 (the first TM of REEP1) and are likely involved in cytl stability"

It is unclear to me how the authors know where exactly the TMs end, especially for TM2 and 3, whose cytosolic ends fall in the middle of $\alpha 3$ and $\alpha 4$, respectively. I do not think the structure prediction softwares can provide this information. The authors should describe how they obtain this information.

Figure 1 legend

"The estimated position of the lipid bilayer is shown as a gray box with its thickness labeled"

This is related to the questions of where the ends of TMs are. How did the authors know the position of the lipid bilayer?

(3)

The method of using 3D-printed model to deduce the structure of oligomers is quite novel and should be better documentd. I searched the literature and could only found the following one paper that has used a similar method to study protein structures. It has a very detailed description of how the 3D-printed models were generated.

Blaber, M. Evaluation of steric entanglement in coiled-coil and domain-swapped protein interfaces using 3D printed models. *J Proteins Proteom* 13, 219–226 (2022).

<https://doi.org/10.1007/s42485-022-00099-w>

In contrast, this manuscripr has only a one-sentence description of the method: "The space-filling model of Yop1p (residues 1-154) is 3D-printed, manually assembled, scanned, and rendered by Maya software." I suggest the authors add more details in the Methods section, including the material used for 3D printing and the type of printer used. In addition, how the models were scanned should be described.

Minor points

(1)

Line 48

"reticulon homolog domain (RHD)" should be "reticulon homology domain (RHD)".

(2)

Line 166

"These practices indicate high confidence in the structural prediction and that membrane curvature in ER tubules is mainly generated and stabilized by curved Yop1p oligomers."

It is unclear to me what "These practices" means. The authors may want to consider rewriting this sentence.

(3)

Line 170

"In the manually assembled TM4-based dimer, we noticed that a pair bulky residue makes extensive contact in the bottom of the TMs (Fig. 2F, bottom view)."

"a pair bulky residue" should be "a pair of bulky residues".

(4)

Line 283

"The TM2 and TM4-mediated dimer formation constitutes the core power of Yop1p RHD assembly."

I think it may be better to say "core driver" instead of "core power".

(5)

Line 286

"whereas the cAPHs stay back-to-back in the TM4 dimer (Fig. 2F)."

It is unclear to me what "stay back-to-back" means. The authors may want to consider rephrasing.

(6)

Line 287

"Collectively, the combination of various dimers leads to oligomerization."

Perhaps changing "combination of various dimers" to "combination of various dimeric interaction modes" will make the sentence read better.

(7)

Title of figure 1

"Structure of REEP proteins" should be "Structures of REEP proteins"

(8)

Legend of Figure 1

"The hydrophobic face of the cAPH is embraced mainly by residues from TM3."

Shouldn't the hydrophobic face of the cAPH be in contact with the lipid? How can it be also embraced by residues from TM3?

(9)

Legend of Figure 1

"(J) As in (A), but with double mutants."

I don't see a panel J in this figure.

REVIEWER COMMENTS

Reviewer #1 (Remarks to the Author):

This manuscript critically examines the mechanistic basis for how reticulons and REEPs induce, maintain, and stabilize the tubular morphology of the ER. The authors use clever strategies of exploiting structures of these proteins from AI-predicted algorithms and in the process validate these algorithms and establish a basic design principle on what constitutes fundamental determinants for these proteins to induce membrane curvature. The manuscript asserts that the contributions of amphipathic helices in these proteins is secondary to their ability to oligomerize and propagate their intrinsic curvatures across a large membrane surface. The results presented in the manuscript are arrived at from rigorous experiments and the description and inferences from primary data are quite valid. Furthermore, assessment of mechanism by which mutations found in hereditary spastic paraplegia better relate not to the intrinsic curvature-associated wedging features of these proteins but to their inability to self-associate into oligomers is quite a significant finding.

We thank the reviewer for finding our work rigorous and significant.

Oligomerisation among REEPs and reticulons have been previously reported to be an important determinant for their membrane curvature-inducing properties. The authors themselves cite these paper in their manuscript (Brady et al. (2015) PNAS, Wang et al. (2021) Nat. Commun.). An obvious consideration therefore is the novelty of the results. Based on the data presented here, this manuscript suggests that the prime determinants for curvature induction lies not in the ability for these proteins to wedge into one monolayer but from their ability to self-associate. This is quite a significant departure from current models. But the present manuscript just lays out these significant findings quite flatly without giving any opportunity for the reader to compare and contrast results presented here with those reported earlier. If the authors indeed believe that the major determinant for curvature induction among reticulons and REEPs is their ability to self-associate then this should be stated clearly and explicitly and would warrant a more fleshed out discussion section emphasising this novel finding.

We appreciate the suggestion. In the revision, we added detailed comparison between models and clearly emphasized the significance and novelty of our findings (**lines 60-64, 303-315, 325-340**).

The term “multifaceted” in the title and text is quite vague and perhaps even misleading. If the authors assert that the dominant mechanism for curvature induction is the self-association among reticulons and REEPs then this should be stated clearly in the manuscript and reflected in the title and the text.

As suggested, we now change the title to “**Oligomeric scaffolding for curvature generation by ER tubule-forming proteins**”, and clearly state our findings in the text.

Experiments based on EGS crosslinking strategy suggest that REEPs and reticulons predominantly exist as dimers. Why would these proteins not form oligomers? This would admittedly be a prerequisite for them to propagate their intrinsic curvature across micron-long ER tubules.

The action of EGS demands solvent-exposing amine groups, which are found mostly in side chains of R and K. Based on our structural prediction, we found that tubule-forming proteins we analyzed have different numbers of non-TM R/Ks: REEP5, 21; REEP1 CTH^Y 14. We thus believe that the weak cross-linking seen in **Supplementary Fig. 5G** is mostly due to limited number of R/Ks. Sufficient trimers, tetramers, or even pentamers were observed by EGS cross-linking with Yop1p, REEP5 and tested RTNs (**PMID: 18309084 and 18442980**). In addition, the probability of forming a high order oligomer is much lower than that of a dimer. It is reasonable that dimer is the strongest band within the oligomeric ladder.

A significant claim from the authors is that the predominant curvature-inducing determinant among REEPs and reticulons is their ability to oligomerize and not so much that they wedge the membrane. Can this be substantiated by an analysis of the intrinsic curvature of the Yop1 and the deltaNT-TM1 Yop1. This can be done by analysing the 3D printed models for their interfacial area to volume ratios. Perhaps the reason why deletion of the NT-TM1 in Yop1 does not compromise its functions lies in the fact that delta NT-TM1 still retains an intrinsic curvature.

We thank the reviewer for this suggestion. We performed the analysis using Yop1p Δ NT-TM1 and found the same results as wild type Yop1p (**Supplementary Fig. 3C-G**). These results nicely explain why deletion of the NT-TM1 does not compromise its functions.

Reconstitution of purified proteins into liposomes uses the E. coli polar lipid extract, which is strange choice of lipids. Could the formation of tubules be attributed to the choice this lipid mix. Were other lipid mixtures tested? The E. coli polar extract contains a higher proportion of PE, which by itself could facilitate the formation of curvature owing to its tendency to exist in the hexagonal phase. The manuscript would benefit from an analysis of reconstitution of proteins in a membrane with a lipid composition that otherwise prefers to remain in the lamellar phase. This could also test the true potential of reticulons to induce curvature.

When we first reported the tubulation assay of purified Yop1p (Science, 2008), we tested a variety of lipid compositions and found that tubule formation by Yop1p is not sensitive to lipid types. To confirm these findings with newly tested REEP1 proteins, we tried yeast polar lipids (YPL) to avoid higher proportion of PE and obtained the same results (**Supplementary Fig. 4F**). Notably, YPL has been used to reconstitute purified REEP1 and REEP5, and consistently, no tubules were observed (**PMID: 28225760**).

Although reported earlier, it's curious that the aberrant ER morphology seen in Yop1/Sey1 deletion strains can be rescued with just the expression of Yop1. Does this mean that membrane fusion by Sey1p is not necessary or that Sey1p functions redundantly with Yop1 as a curvature inducing protein?

It has been reported that the yeast ER fusogen Sey1p can be backed up by ER SNARE Ufe1p (**PMID: 22508509**), which explained why unlike ATL, Sey1p deletion does not cause lethality and evident changes in ER morphology. The curvature generation capacity, mostly by having a transmembrane hairpin (TMH), is weaker with Sey1p than its *Drosophila* homolog (discussed in **PMID: 28225760**). In short, it is not clear to us why the double deletion has the same ER phenotype as the Rtn1p/Rtn2p/Yop1p triple deletion. Nevertheless, we eliminated the potential complication with Sey1p by repeating the experiments in TKO when comparing REEPs (**Supplementary Fig. 5B**).

Reviewer #2 (Remarks to the Author):

In this manuscript, the authors used RoseTTAFold, AlphaFold, and 3D-printed models to predict the structures of yeast Yop1p and related human REEP proteins. Structure prediction results and crosslinking analysis data together indicate that there exist two different homotypic interaction modes, with TM2 and TM4 forming the interaction interface, respectively. Curved-shaped tetramers can be formed by alternately using these two interaction modes. These findings provide novel insights into how these proteins generating membrane curvature and forms ER tubules.

I have the following comments and suggestions for the authors to consider.

Major points:

(1)

The Brady et al. 2015 PNAS paper proposed that Yop1p contains four hydrophobic helices long enough to span the ER bilayer fully (see Figure 8 of that paper).

In Figure 1B, the TMs of Yop1 are shown as not spanning the membrane bilayer.

The authors should explain why they choose to show the TMs this way and discuss why they do not agree with the model of Brady et al.

We thank the reviewer for pointing out this discrepancy. As illustrated below, our TMs are indeed shorter than previously proposed. The start of TM1 and the end of TM4 are determined by apparent transition from APH to TM helix. The minor difference seen between the two models lies mostly in the connecting loop, which does alter TM length. The Brady model counted these residues as part of the TM helices, making their TMs longer than those in our model. The end of TM2 and the start of TM3 are determined by 1) appearance or disappearance of polar residues, and 2) cross-comparison with TM1 and TM4 and between TM2 and TM3 to make sure that all TMs start or end at the same altitude. These considerations render shorter TMs when compared to the previous model. In addition, our model predicts that TMs are in general tilted and interlocked, making them less likely spanning the entire bilayer. We have added this point in the text (lines 83-85, 90-91).

(2)

Line 93

"Notably, both APHs lie on a plane that aligns with the cytosolic ends of all TMs (Fig. 1B)."

Line 265

"P19R and A20E localize to the end of TM2 (the first TM of REEP1) and are likely involved in cytL stability"

It is unclear to me how the authors know where exactly the TMs end, especially for TM2 and 3, whose cytosolic ends fall in the middle of α_3 and α_4 , respectively. I do not think the structure prediction softwares can provide this information. The authors should describe how they obtain this information.

In our predicted model, TM1 and TM4 are connected to nAPH and cAPH, respectively. The trace of these APHs are almost perpendicular to that of the TMs. Thus, it is reasonable to

speculate that TM1 starts right after the turn and TM4 ends right before the turn. If the start of TM1 and the end of TM4 marks the surface of the membrane, then the two APHs would lie on this surface, with the hydrophobic face dipping into the lipid bilayer and the hydrophilic face exposing to cytosol. As mentioned above, the end of TM2 and the start of TM3 are determined by 1) appearance or disappearance of polar residues, and 2) cross-comparison with TM1 and TM4 and between TM2 and TM3 to make sure that all TMs start or end at the same altitude. We now describe these considerations in the text (**lines 83-85**).

Figure 1 legend

"The estimated position of the lipid bilayer is shown as a gray box with its thickness labeled" This is related to the questions of where the ends of TMs are. How did the authors know the position of the lipid bilayer?

We drew the gray box (assuming a lipid bilayer is 3.5 nm in thickness) proportionally to the size of the protein as measured in PyMoL. The determination of the ends of TMs are described above. Finally, we aligned the top of the box with the start/end of the TMs. By doing so, we predicted that the TMs of Yop1p would less likely span the lipid bilayer.

(3)

The method of using 3D-printed model to deduce the structure of oligomers is quite novel and should be better documented. I searched the literature and could only found the following one paper that has used a similar method to study protein structures. It has a very detailed description of how the 3D-printed models were generated.

Blaber, M. Evaluation of steric entanglement in coiled-coil and domain-swapped protein interfaces using 3D printed models. *J Proteins Proteom* 13, 219–226

(2022). <https://doi.org/10.1007/s42485-022-00099-w>

In contrast, this manuscript has only a one-sentence description of the method: "The space-filling model of Yop1p (residues 1-154) is 3D-printed, manually assembled, scanned, and rendered by Maya software." I suggest the authors add more details in the Methods section, including the material used for 3D printing and the type of printer used. In addition, how the models were scanned should be described.

We thank the reviewer for the note. The mentioned paper would like to simulate a domain swap event by using 3D-printed model. Therefore, they chose relatively flexible materials for the purpose. We, on the other hand, predict the rigidity of the monomer and thus chose the conventional resin for printing. As suggested, we now include detailed description in the Methods section (**lines 353-358**).

Minor points

(1)

Line 48

"reticulon homolog domain (RHD)" should be "reticulon homology domain (RHD)".

Corrected.

(2)

Line 166

"These practices indicate high confidence in the structural prediction and that membrane curvature in ER tubules is mainly generated and stabilized by curved Yop1p oligomers."

It is unclear to me what "These practices" means. The authors may want to consider rewriting this sentence.

Rephrased.

(3)

Line 170

"In the manually assembled TM4-based dimer, we noticed that a pair bulky residue makes extensive contact in the bottom of the TMs (Fig. 2F, bottom view)."

"a pair bulky residue" should be "a pair of bulky residues".

Corrected.

(4)

Line 283

"The TM2 and TM4-mediated dimer formation constitutes the core power of Yop1p RHD assembly."

I think it may be better to say "core driver" instead of "core power".

Rephrased.

(5)

Line 286

"whereas the cAPHs stay back-to-back in the TM4 dimer (Fig. 2F)."

It is unclear to me what "stay back-to-back" means. The authors may want to consider rephrasing.

Rephrased.

(6)

Line 287

"Collectively, the combination of various dimers leads to oligomerization."

Perhaps changing "combination of various dimers" to "combination of various dimeric interaction modes" will make the sentence read better.

Rephrased.

(7)

Title of figure 1

"Structure of REEP proteins" should be "Structures of REEP proteins"

Corrected.

(8)

Legend of Figure 1

"The hydrophobic face of the cAPH is embraced mainly by residues from TM3."

Shouldn't the hydrophobic face of the cAPH be in contact with the lipid? How can it be also embraced by residues from TM3?

We meant the side of the hydrophobic face. We have rephrased the sentence accordingly.

(9)

Legend of Figure 1

"(J) As in (A), but with double mutants."

I don't see a panel J in this figure.

Corrected.

REVIEWERS' COMMENTS

Reviewer #1 (Remarks to the Author):

The revised manuscript and the author's response adequately addresses my comments. I congratulate the authors on this wonderful piece of work.

Thomas Pucadyil

Reviewer #2 (Remarks to the Author):

The revision has fully addressed my concerns. I support the publication of this paper.